# Noise Consistency Regularization for Improved Subject-Driven Image Synthesis

Yao Ni[1,†]    Song Wen[2]    Piotr Koniusz[1,3]    Anoop Cherian[4,*]

[1]The Australian National University    [2]Rutgers University    [3]Data61♥CSIRO
[4]Mitsubishi Electric Research Laboratories (MERL)

[1]yao.ni@anu.edu.au [2]song.wen@rutgers.edu [3]piotr.koniusz@data61.csiro.au [4]cherian@merl.com

## Abstract

*Fine-tuning Stable Diffusion enables subject-driven image synthesis by adapting the model to generate images containing specific subjects. However, existing fine-tuning methods suffer from two key issues: underfitting, where the model fails to reliably capture subject identity, and overfitting, where it memorizes the subject image and reduces background diversity. To address these challenges, we propose two auxiliary consistency losses for diffusion fine-tuning. First, a prior consistency regularization loss ensures that the predicted diffusion noise for prior (non-subject) images remains consistent with that of the pre-trained model, improving fidelity. Second, a subject consistency regularization loss enhances the fine-tuned model's robustness to multiplicative noise modulated latent code, helping to preserve subject identity while improving diversity. Our experimental results demonstrate that incorporating these losses into fine-tuning not only preserves subject identity but also enhances image diversity, outperforming DreamBooth in terms of CLIP scores, background variation, and overall visual quality.*

## 1. Introduction

Recent advancements in deep learning have significantly enhanced generative modeling capabilities, driven by the availability of large-scale datasets such as LAION-5B [51] and ImageNet [10]. Among various generative architectures, including VAEs [23] and GANs [17, 33], diffusion models [42, 43, 49] have emerged as state-of-the-art, demonstrating remarkable performance in image synthesis [2, 6, 30, 36], video generation [3, 4], and 3D content creation [37, 41, 58]. Notably, Stable Diffusion [45] has gained widespread adoption as a foundational model for solving various downstream generative tasks through fine-tuning. Applications include subject-driven image personalization [25, 26, 47] and controlled image synthesis, as ex-emplified by ControlNet [21, 64, 67].

Subject-driven personalization [25, 26, 47] aims to adapt pre-trained generative models to synthesize images of specific subjects while preserving both subject identity and the model's generalization ability for diverse image generation. A common approach is to fine-tune diffusion models on a limited set of subject images using LoRA [20], while leveraging linguistic prompts to encode subject concepts [39]. For example, fine-tuning with the prompt "*A photo of a* $[V]$ *dog*" allows Stable Diffusion to associate the subject with its broader class ("dog"), facilitating contextual understanding for more diverse synthesis and direct text control. An example is shown in Figure 1.

However, fine-tuning poses a major challenge: catastrophic forgetting [25, 47], where the model loses its pre-trained knowledge. To counter this, DreamBooth [47] introduces a prior preservation loss, generating a set of (e.g., 100–200 images) class-specific images (e.g., "*A photo of a dog*") using the original model and enforcing consistency during fine-tuning. While this helps retain prior knowledge, empirical results show that it often fails to fully preserve subject identity and reduces image diversity, limiting its effectiveness [7, 52, 61].

To tackle these issues, we introduce novel auxiliary consistency losses for fine-tuning Stable Diffusion, aiming to preserve subject identity while improving image fidelity and diversity. We argue that LoRA, which adapts Stable Diffusion for subject images, does not need to re-learn class prior images, as the pre-trained model already captures the class concept. We also observe that fine-tuning LoRA on ground-truth noise for prior images lowers fidelity by causing the model to forget the realism learned by pre-trained.

To address this, we propose feeding prior images to both the fine-tuned and pre-trained models, enforcing consistency between the noise predicted by both models. For subject images, we maintain the standard approach where the fine-tuned model predicts the ground-truth noise added to its latent code. This approach improves the fidelity of generated subject images by retaining the realism developed in the pre-trained model while allowing LoRA to focus its ca-

---

*The corresponding author.    †Work done during internship at MERL.

pacity on fitting the subject images.

While this prior consistency regularization preserves identity and improves fidelity, we find that it reduces diversity in the generated images due to the limited number of subject images. To enhance diversity, we introduce multiplicative Gaussian noise modulation [32, 35] to the latent vectors, which diversifies the pattern of subject images without altering their semantics. We then enforce consistency between the noised and clean latent codes. Combining these approaches improves background diversity while ensuring high fidelity and preserving subject identity.

To validate the effectiveness of our proposed regularizations, we conduct experiments on the benchmark dataset from [41], which includes 30 subject classes spanning both live subjects and objects. Our method achieves state-of-the-art performance in subject-driven text-to-image synthesis, as measured by CLIP [39] and DINO [5] scores. Additionally, We demonstrate the generality of our approach by integrating it into both DreamBooth [47] and DCO [26], yielding clear improvements. Qualitative results further highlight the diversity of generated images while preserving subject identity, reinforcing the effectiveness of our method.

Before we proceed with describing our technical details, we summarize the key contributions of this paper below:

- We address the important issue of preserving subject identity while avoiding overfitting in subject-driven Stable Diffusion-based image synthesis.
- To preserve identity, we propose enforcing a LoRA-based fine-tuning model with an auxiliary noise consistency regularization using prior synthesized images.
- To improve the diversity of synthesized images, we propose adding multiplicative noise to the subject images and enforcing the fine-tuned Stable Diffusion model to be consistent with both the noised and clean latent codes.
- We present experiments demonstrating that our method can improve the fidelity of the subject image while also enhancing diversity compared to previous methods.

## 2. Related work

**Subject-driven personalization**. These approaches [25, 26, 28, 47, 48, 54] can be broadly categorized into three broad directions: full model fine-tuning, partial model fine-tuning, and embedding optimization. In full model fine-tuning, DreamBooth [47] pioneered the idea of fine-tuning all model weights to encode subjects as unique identifiers, enabling the generation of the learned subjects in various contexts and styles. HyperDreambooth [48] uses a hypernetwork that can efficiently produce a small set of personalized weights from a single image of a person. However, this comprehensive fine-tuning approach often faces challenges in computational efficiency and overfitting. To address these limitations, several works have explored partial model fine-tuning. CustomDiffusion [25] demonstrated

that fine-tuning only the cross-attention layers is sufficient for concept learning while achieving superior performance in multi-concept compositional generation. StyleDrop [54] enables to synthesize an image following a specific style while only fine-tuning 1% of total model parameters. Similarly, DCO [26] proposed a fine-tuning method that preserves the pre-trained model's compositional capabilities through implicit reward models. An alternative approach focuses on embedding optimization. Textual Inversion [14] proposed optimizing word embeddings while keeping the pre-trained diffusion model frozen, effectively capturing unique concepts with minimal parameter updates. Building upon this, hybrid approaches [1, 16, 59] have emerged that combine both word embedding optimization and selective model weight updates. Another notable work in this direction [13] further explored gradient-based embedding optimization techniques, while [31] focused on zero-shot image-conditioned text-to-image synthesis. In contrast to these approaches, we consider consistency losses with a focus in preserving the subject identity and prior knowledge of diffusion models during fine-tuning, making our approach applicable to any prior fine-tuning technique.

**Parameter-efficient fine-tuning.** The emergence of large-scale foundation models has introduced significant challenges for fine-tuning on limited datasets due to their massive parameter counts. To address the efficiency and overfitting concerns inherent in traditional fine-tuning approaches, Parameter Efficient Fine-Tuning (PEFT) methods have emerged as a promising solution. PEFT techniques can be categorized into several major approaches. Adapter tuning [8, 22, 44] introduces trainable layers within the pre-trained model architecture, creating efficient bottlenecks for adaptation. Soft prompt tuning [15, 55, 57, 59] is an efficient technique for adapting large foundation models to specific tasks by learning task-relevant prompt embeddings, which guide model outputs without modifying the core model weights. Low-Rank Adaptation (LoRA) [11, 19, 20, 24, 65, 68] has gained particular prominence by decomposing weight updates into products of low-rank matrices, dramatically reducing the parameter count while maintaining performance. Recent innovations have explored alternative decomposition strategies. SVDiff [18] employs singular value decomposition of pre-trained weights, focusing adaptation on singular values alone. OFT [27, 38] preserves model structure by incorporating trainable orthogonal matrices while maintaining hyperspherical energy. KronA [12, 29] introduces a novel approach using Kronecker products of smaller matrices to construct weight updates efficiently. SODA [66] decomposes the pretrained weight matrix and fine-tunes its individual components. Our proposed consistency losses attempt to balance PEFT weight adaption while minimizing its impact on the prior knowledge already learned in the pre-trained model.

**Consistency regularization.** The concept of consistency regularization has been explored periodically in various contexts. FixMatch [53] applies it to augmented images for semi-supervised learning, while OpenMatch [50] leverages it for outlier predictions in open-set semi-supervised learning. R-Drop [60] introduces consistency regularization in transformers [56] with dropout for NLP tasks. In GAN training, CR [63] enforces consistency between augmented real and fake images, and CAGAN [34] applies it to discriminators with dropout. Sem-GAN [9] enforces semantic consistency for image-to-image translation. Despite the empirical success of consistency regularization, theoretical analysis has been limited. NICE [32] addresses this gap by showing that consistency regularization reduces latent feature gradients, stabilizing GAN training. PACE [35] further connects consistency regularization to improved generalization in deep learning. Additionally, consistency regularization has been applied to fine-tuned and pre-trained models [35, 40, 46] to retain pre-trained knowledge. However, its potential for preserving subject identity while enhancing background diversity remains unexplored. Our method addresses this gap, offering a novel approach that improves both subject consistency and background diversity, providing a compelling advancement over previous methods.

# 3. Background

Before detailing our approach, we introduce key background concepts in deep generative modeling that underpin our contributions. This section covers the fundamental principles of Stable Diffusion and subject-driven image synthesis while establishing our notation.

## 3.1. Stable Diffusion

A breakthrough in deep generative modeling happened with the introduction of the Stable Diffusion model, a text-to-image model pretrained on extensive text-image pairs $\{(\boldsymbol{p}, \boldsymbol{x})\}$, with each pair linking a text prompt $\boldsymbol{p}$ to its corresponding image $\boldsymbol{x}$. Such a paired dataset forms the foundation for training the model's architecture, which includes an auto-encoder with encoder $\mathcal{E}(\cdot)$ and decoder $\mathcal{D}(\cdot)$, a CLIP text encoder $\tau(\cdot)$, and a UNet-based conditional diffusion model $f(\cdot)$. Within this system, the encoder transforms images $\boldsymbol{x}$ into latent representations $\boldsymbol{z} = \mathcal{E}(\boldsymbol{x})$, creating compact codes that enable the decoder to effectively reconstruct images as needed. Operating within this latent space rather than raw pixels, the diffusion process introduces noise at time step $t$ with $\boldsymbol{\epsilon} \sim \mathcal{N}(\mathbf{0}, \boldsymbol{I})$, producing noisy latent code $\boldsymbol{z}_t = \alpha_t \boldsymbol{z} + \beta_t \boldsymbol{\epsilon}$ controlled by coefficients $\alpha_t$ and $\beta_t$ that manage the denoising schedule. These noisy representations drive the training of the conditional diffusion model $f(\cdot)$ through the objective:

$$\min \mathbb{E}_{\boldsymbol{p}, \boldsymbol{z}, \boldsymbol{\epsilon}, t} \left[ \|\boldsymbol{\epsilon} - f(\boldsymbol{z}_t, t, \tau(\boldsymbol{p}))\|_2^2 \right]. \quad (1)$$

The noise prediction objective, guided by text embedding $\tau(\boldsymbol{p})$ and conditioned on time step $t$, enables diffusion models to iteratively refine noisy latent codes $\boldsymbol{z}_t$. This refinement process, enhanced by an expressive transformer architecture and knowledge derived from diverse training data, endows Stable Diffusion model with robust semantic priors. Leveraging these priors, the model generates high-quality and diverse images directly from natural language prompts while ensuring precise alignment between textual input and visual output, delivering reliable text-to-image generation.

## 3.2. Subject-driven Synthesis

While Stable Diffusion models excel at general text-to-image synthesis, they struggle to faithfully replicate specific subjects from a reference set or synthesize these subjects in new contexts. To overcome this limitation, subject-driven fine-tuning extends the model's vision-language dictionary by binding a new token $[V]$ to user-defined subjects.

Let $s$ represent a subject represented by a small set of subject images $\mathcal{S} = \{\boldsymbol{x}_i\}_{i=1}^{K}$, where the number of available images $K$ is very small. To expand the Stable Diffusion vision-language dictionary, subject-driven synthesis fine-tunes the model using a subject-specific prompt $\boldsymbol{p}_s$, e.g., *"a photo of $[V]$ dog"* and optimize the objective:

$$\min_{\Delta\boldsymbol{\theta}} \mathcal{L}_{\mathrm{s}} := \mathbb{E}_{\boldsymbol{x} \sim \mathcal{S}, \boldsymbol{z} = \mathcal{E}(\boldsymbol{x}), \boldsymbol{\epsilon}, t} [\|\boldsymbol{\epsilon} - f_{\Delta\boldsymbol{\theta}}(\boldsymbol{z}_t, t, \tau(\boldsymbol{p}_s))\|_2^2], \quad (2)$$

where $\Delta\boldsymbol{\theta}$ denotes the trainable LoRA weights added to the otherwise frozen pre-trained model, resulting in the fine-tuned model $f_{\Delta\boldsymbol{\theta}}$.

The naïve fine-tuning scheme in Eq. (2) aligns the generated image space of Stable Diffusion with subject images. However, this alignment, when applied to a limited number of subject images, risks overfitting to these images, discarding pre-trained knowledge in the model, thereby reducing output diversity. To preserve diversity while retaining pre-trained knowledge, DreamBooth [47] introduces prior preservation, generating a set of class images $\mathcal{P} = \{\boldsymbol{x}_i\}_{i=1}^{K_p}$ (where $K_p$, e.g., 100–200 is the number of prior images) using a generic prompt $\boldsymbol{p}_p$ such as *"a photo of dog"*. The model is then fine-tuned using a combined objective:

$$\min_{\Delta\boldsymbol{\theta}} \mathcal{L}_{\mathrm{prior}} := \mathbb{E}_{\boldsymbol{x} \sim \mathcal{P}, \boldsymbol{z} = \mathcal{E}(\boldsymbol{x}), \boldsymbol{\epsilon}, t} [\|\boldsymbol{\epsilon} - f_{\Delta\boldsymbol{\theta}}(\boldsymbol{z}_t, t, \tau(\boldsymbol{p}_p))\|_2^2],$$

$$\min_{\Delta\boldsymbol{\theta}} \mathcal{L}_{\mathrm{dreambooth}} := \mathcal{L}_{\mathrm{s}} + \lambda_{\mathrm{prior}} \mathcal{L}_{\mathrm{prior}}, \quad (3)$$

where $\lambda_{\mathrm{prior}}$ regulates the balance between subject fidelity and pre-trained knowledge retention.

# 4. Proposed Approach

While DreamBooth effectively integrates new concepts into Stable Diffusion, its emphasis on optimizing performance for a limited set of subject and class prior images introduces

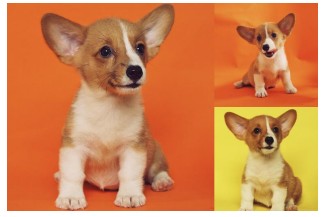 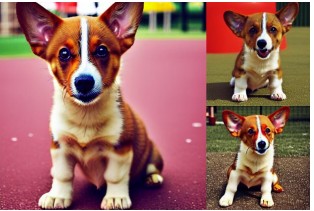 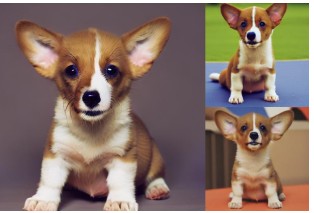 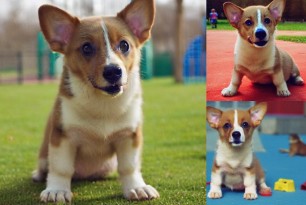

| (a) Input real subject images. | (b) DreamBooth w/ $\mathcal{L}_s$ and $\mathcal{L}_{prior}$. | (c) Applying $\mathcal{L}_s$ and $\mathcal{L}_{cp}$. | (d) Applying $\mathcal{L}_s$, $\mathcal{L}_{cp}$ and $\mathcal{L}_{cs}$ |

Figure 1. Generated images using different methods with the prompt *"A photo of a [V] dog on the playground"*. DreamBooth reduces fidelity and loses subject identity. Applying $\mathcal{L}_{cp}$ improves fidelity but reduces background diversity, while $\mathcal{L}_{cs}$ enhances it.

critical flaws. The combination of scarce training data and the model's vast parameter space makes it highly prone to overfitting, distorting its original representation of natural image distributions. This problem is further exacerbated by a fundamental contradiction in the training process, where the ground-truth noise added to class images differs from the noise the pretrained model naturally predicts for those inputs. Simultaneously, forcing the LoRA weights $\Delta\boldsymbol{\theta}$ to adapt to both subject and prior images introduces redundancy, as the pretrained model already encodes prior semantics. This mismatch skews the latent diffusion space toward the prior images, corrupting the model's learned data manifold during pre-training. Consequently, the susceptibility to overfitting and the noise-prediction mismatch degrades its ability to generate diverse and realistic outputs that preserve subject identity and lose background details. This problem is illustrated in Figure 1, where for the subject images in Figure 1a, DreamBooth is seen to lose realism and subject identity as shown in Figure 1b.

## 4.1. Consistency Regularization for Fidelity

To resolve the noise-prediction mismatch and eliminate redundant adaptation to prior images, we propose enforcing consistency between the fine-tuned and pretrained models for prior images $\mathcal{P}$. Specifically, we input diffusion-noised prior images into both the fine-tuned and pre-trained models and optimize the LoRA weights $\Delta\boldsymbol{\theta}$ in the fine-tuned model to ensure that the predicted noise remains consistent with the prediction from the pre-trained model. Our objective can be mathematically stated as:

$$\min_{\Delta\boldsymbol{\theta}} \mathcal{L}_{cp} := \tag{4}$$
$$\mathbb{E}_{\boldsymbol{x}\sim\mathcal{P},\boldsymbol{z}^p=\mathcal{E}(\boldsymbol{x}),\boldsymbol{\epsilon},t}[\|f_{\Delta\boldsymbol{\theta}}(\boldsymbol{z}_t^p, t, \tau(\boldsymbol{p}_p)) - f_{ref}(\boldsymbol{z}_t^p, t, \tau(\boldsymbol{p}_p))\|_2^2],$$

where $f_{ref}$ denotes the pretrained conditional latent Stable Diffusion model. By enforcing consistency between the fine-tuned and pretrained models on prior images, the fine-tuned model inherently retains the pre-trained knowledge, eliminating the need to relearn prior images. This allows the fine-tuned model to fully allocate the LoRA weights $\Delta\boldsymbol{\theta}$ to learning the subject identity, ensuring that the fine-tuned

model synthesizes the subject faithfully while preserving the natural and realistic outputs inherited from $f_{ref}$.

## 4.2. Consistency Regularization for Diversity

While the consistency regularization on prior images effectively preserves subject identity and fidelity, synthesizing diverse outputs remains challenging due to the limited number of subject images. As illustrated in Figure 1c, combining $\mathcal{L}_s$ and $\mathcal{L}_{cp}$ improves the realism and preserves the subject identity compared to DreamBooth. However, this comes at the cost of reduced background diversity.

To address this, we propose multiplicative noise modulation, a technique inspired by its success in stabilizing GAN training [32] and improving generalization in parameter-efficient fine-tuning [35], to expand the latent space of subject images without altering the core image semantics. For each subject image, we perturb its latent code as:

$$\boldsymbol{z}' = \boldsymbol{z} \odot \boldsymbol{\epsilon}_m, \quad \boldsymbol{\epsilon}_m \sim \mathcal{N}(\boldsymbol{1}, \sigma^2\boldsymbol{I}), \tag{5}$$

where $\sigma^2$ is the variance of the multiplicative noise. Notably, while multiplicative noise has been applied in the prior works [32, 35], its use to enhance diversity in diffusion models has never been explored. This semantics-preserving perturbation introduces controlled variations into the latent space, diversifying outputs without distorting the subject's identity.

To ensure noise-augmented latents retain subject identity while improving diversity, we enforce consistency between predictions for clean latent code $\boldsymbol{z}$ and noise-modulated latent codes $\boldsymbol{z}'$ of the subject images via:

$$\min_{\Delta\boldsymbol{\theta}} \mathcal{L}_{cs} := \tag{6}$$
$$\mathbb{E}_{\boldsymbol{x}\sim\mathcal{S},\boldsymbol{z}=\mathcal{E}(\boldsymbol{x}),\boldsymbol{z}',\boldsymbol{\epsilon},t}[\|f_{\Delta\boldsymbol{\theta}}(\boldsymbol{z}_t, t, \tau(\boldsymbol{p}_s)) - f_{\Delta\boldsymbol{\theta}}(\boldsymbol{z}_t', t, \tau(\boldsymbol{p}_s))\|_2^2].$$

Note that $\boldsymbol{z}_t = \alpha_t\boldsymbol{z} + \beta_t\boldsymbol{\epsilon}$ and $\boldsymbol{z}_t' = \alpha_t(\boldsymbol{z}\odot\boldsymbol{\epsilon}_m) + \beta_t\boldsymbol{\epsilon}$ share the same diffusion noise $\boldsymbol{\epsilon}$ and time step $t$.

## 4.3. Overall Objective

With consistency regularization applied to both prior and subject images, our final objective for robust subject-driven

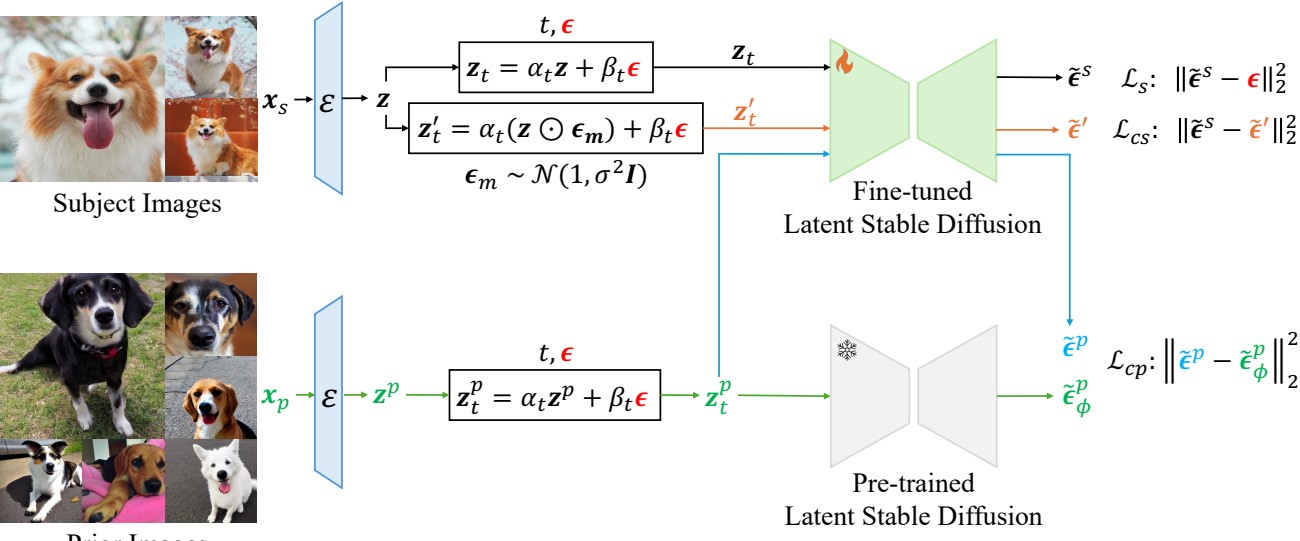

Figure 2. Our pipeline applies two forms of consistency regularization: one over subject images and one over prior images. For the subject images, we modulate the latent code $z$ with multiplicative noise $\epsilon_m \sim \mathcal{N}(1, \sigma^2 I)$, then add identical diffusion noise $\epsilon$ to both the clean and noise-modulated latent codes, yielding $z_t$ and $z_t'$. These are fed into the fine-tuned Stable Diffusion model. This model has two objectives for subject images: (1) $\mathcal{L}_s$, where the predicted noise $\tilde{\epsilon}^s$ approximates the ground truth noise $\epsilon$, and (2) $\mathcal{L}_{cs}$, where the prediction $\tilde{\epsilon}'$ should be consistent with $\tilde{\epsilon}^s$. For prior images, we input the diffused latent code $z_t^p$ into both the fine-tuned and pre-trained latent diffusion models, enforcing consistency between the fine-tuned model's prediction $\tilde{\epsilon}^p$ and the pre-trained model's prediction $\tilde{\epsilon}_\phi^p$.

fine-tuning is formulated as:

$$\min_{\Delta\theta} \mathcal{L} := \mathcal{L}_s + \lambda_{cp}\mathcal{L}_{cp} + \lambda_{cs}\mathcal{L}_{cs}, \qquad (7)$$

where $\lambda_{cp}$ and $\lambda_{cs}$ control the regularization strengths for prior and subject images, respectively.

Our pipeline for implementing these two consistency regularizations is illustrated in Figure 2. For subject images, we apply multiplicative noise modulation to their latent codes and process both the clean and noised codes through the diffusion process. The resulting codes are then fed into the fine-tuned diffusion model, where $\mathcal{L}_s$ predicts the ground-truth noise and $\mathcal{L}_{cs}$ enforces consistency between predictions of both clean and noised codes. For prior images, we apply the diffusion process to their latent codes, which are then fed into both the fine-tuned and pre-trained models, followed by enforcing consistency between their outputs. These two consistency losses strengthen the fine-tuning process, enhancing model robustness and enabling the fine-tuned diffusion model to generate diverse and realistic subject images, as shown in Figure 1d.

# 5. Experiments

In this section, we provide experiments demonstrating the benefits of our proposed regulization schemes. To this end, we fine-tune Stable Diffusion V1.5 [45] on subject images for subject-driven text-to-image synthesis. We qualitatively and quantitatively evaluate our methods and conduct ablation studies on our proposed regularization designs.

## 5.1. Dataset

We use the publicly available dataset from [47]. This dataset consists of 30 subjects and comprising a diverse range of 9 unique live subjects, including animals such as dogs and cats, along with 21 distinct objects. These objects span various categories, such as backpacks, stuffed animals, sunglasses, cartoons, toys, sneakers, teapots, vases, and more. This selection of subjects and objects provides a broad and varied basis for our study.

## 5.2. Evaluation

For each subject, we generate images guided by 10 distinct prompts using the fine-tuned Stable Diffusion model. For each prompt, we synthesize 4 images, resulting in a total of 40 images per subject. We then calculate the CLIP-I, CLIP-T, and DINO scores as follows:

**CLIP-I Score:** We use the CLIP image encoder [39] to obtain embeddings for both the generated images and the real images. We calculate the mean pairwise cosine similarity between embeddings of generated and real images.

**CLIP-T Score:** We input each prompt into the CLIP text encoder to obtain prompt embeddings, then calculate the mean cosine similarity between the prompt embedding and the embeddings of the generated images.

Table 1. Results for subject-driven text-to-image synthesis. We compare our method with DreamBooth [47] and DCO [26], evaluating the performance using three metrics: CLIP-I, CLIP-T, and DINO scores. The reported results represent the average performance across 30 subjects, encompassing both live subjects and object categories. Our method demonstrates superior performance.

| Method | DINO ↑ | CLIP-I ↑ | CLIP-T ↑ |
|---|---|---|---|
| Real Images | 0.703 | 0.864 | − |
| DreamBooth | 0.602 | 0.778 | **0.329** |
| Ours | 0.**634** | **0.792** | 0.324 |
| DreamBooth + DCO | 0.638 | 0.795 | 0.309 |
| Ours + DCO | **0.652** | **0.811** | **0.310** |

**DINO Score:** Using the DINO model [5], we encode the generated and real images and calculate the mean pairwise cosine similarity between their embeddings.

For each method, we report the average of these scores across all 30 subjects, providing a comprehensive assessment of performance across different subjects and prompts.

### 5.3. Hyperparameter settings

We compare our method against DreamBooth [47] and DCO [26] under consistent experimental settings to ensure a fair evaluation. Specifically, we fix the learning rate at 5e-4 and use LoRA with a rank of 4 to fine-tune the UNet of the latent Stable Diffusion model across all methods.

For DreamBooth, we set $\lambda_{prior} = 1$ and generate 100 prior images to preserve prior knowledge, as recommended in the original paper. We also follow the guidelines provided by DCO and set its parameter $\beta$ to 1000, as recommended.

In our method, we maintain $\lambda_{cp} = \lambda_{cs} = 0.5$ for consistency regularization and fix the noise modulation parameter $\sigma = 0.2$. These consistent configurations ensure a robust and meaningful comparison across all methods.

### 5.4. Quantitative Results

In Table 1, we present a comparison of our method with DreamBooth [47] and DCO [26]. Our approach achieves higher CLIP-I and DINO scores than both DreamBooth and DCO, while maintaining a competitive CLIP-T score. These results highlight the effectiveness of our method in producing subject-driven text-to-image generations that capture both visual and semantic alignment, demonstrating a clear advantage over existing techniques.

### 5.5. Qualitative Results

We provide a qualitative comparison of our method with DreamBooth, as shown in Figure 3. From the figure, it is evident that our method excels in preserving the subject's identity, maintaining consistent and recognizable features across the generated images. In contrast, the baseline

method, DreamBooth, struggles to retain the subject's identity, leading to noticeable distortions and inconsistencies. These results clearly demonstrate the superior performance of our approach in subject-driven text-to-image synthesis, underscoring its ability to achieve higher fidelity and visual coherence compared to the baseline.

### 5.6. Ablation studies

Below, we present experiments to justifying $\mathcal{L}_{cp}$ and $\mathcal{L}_{cs}$ improve fidelity and background diversity.

**Impact of $\lambda_{cp}$.** To investigate the impact of $\lambda_{cp}$ on performance, we fixed other hyperparameters and varied $\lambda_{cp}$ from 0.0 to 1.0. The generated images for different values of $\lambda_{cp}$ using the prompt *"A photo of a [V] dog on the playground"* are shown in Figure 4. Comparing images generated with $\lambda_{cp} = 0.0$ and $\lambda_{cp} = 0.2$, we observe that introducing consistency regularization for class prior images through $\mathcal{L}_{cp}$ effectively improves both fidelity and background diversity, confirming its importance and effectiveness. The best fidelity and diversity are achieved at $\lambda_{cp} = 0.5$, with larger values showing minimal visual differences.

**Impact of $\lambda_{cs}$.** We examine the effect of $\lambda_{cs}$ on performance. Keeping other hyperparameters fixed, we varied $\lambda_{cs}$ from 0.0 to 0.8. The generated images for different values of $\lambda_{cs}$ using the prompt *""A photo of a [V] dog on the playground"* are presented in Figure 5. Comparing the images generated with $\lambda_{cs} = 0.0$ and $\lambda_{cs} = 0.2$, we find that introducing consistency regularization for the subject via $\mathcal{L}_{cs}$ effectively enhances background diversity, validating its effectiveness. The best balance between fidelity and diversity is achieved at $\lambda_{cs} = 0.5$, while higher values increase background diversity at the cost of subject identity preservation.

**How does $\mathcal{L}_{cs}$ improve background diversity?** To clarify why the proposed consistency regularization through multiplicative noise modulation enhances the background diversity, we analyze the latent code of these images using the encoder $\mathcal{E}(\cdot)$. Specifically, we generate images by applying different loss combinations, including DreamBooth losses, $\mathcal{L}_s/\mathcal{L}_{cp}$, $\mathcal{L}_s/\mathcal{L}_{cs}$, and $\mathcal{L}_s/\mathcal{L}_{cp}/\mathcal{L}_{cs}$, for the subject dataset shown in Figure 1a. We then compute and plot the histogram of the latent code values. Additionally, we calculate the KL divergence between the latent code of generated images and the prior images, as shown in Figure 6.

Among these configurations, applying $\mathcal{L}_{cs}$ yields the lowest KL divergence with the prior images, indicating that multiplicative noise modulation effectively diversifies the latent code distribution and enhances the background diversity of the generated images. Moreover, applying $\mathcal{L}_{cp}$ alone improves fidelity but reduces diversity, resulting in the highest KL divergence. In contrast, combining $\mathcal{L}_{cs}$ with $\mathcal{L}_{cp}$ effectively mitigates this loss of diversity while maintaining fidelity, achieving a lower KL divergence than Dream-

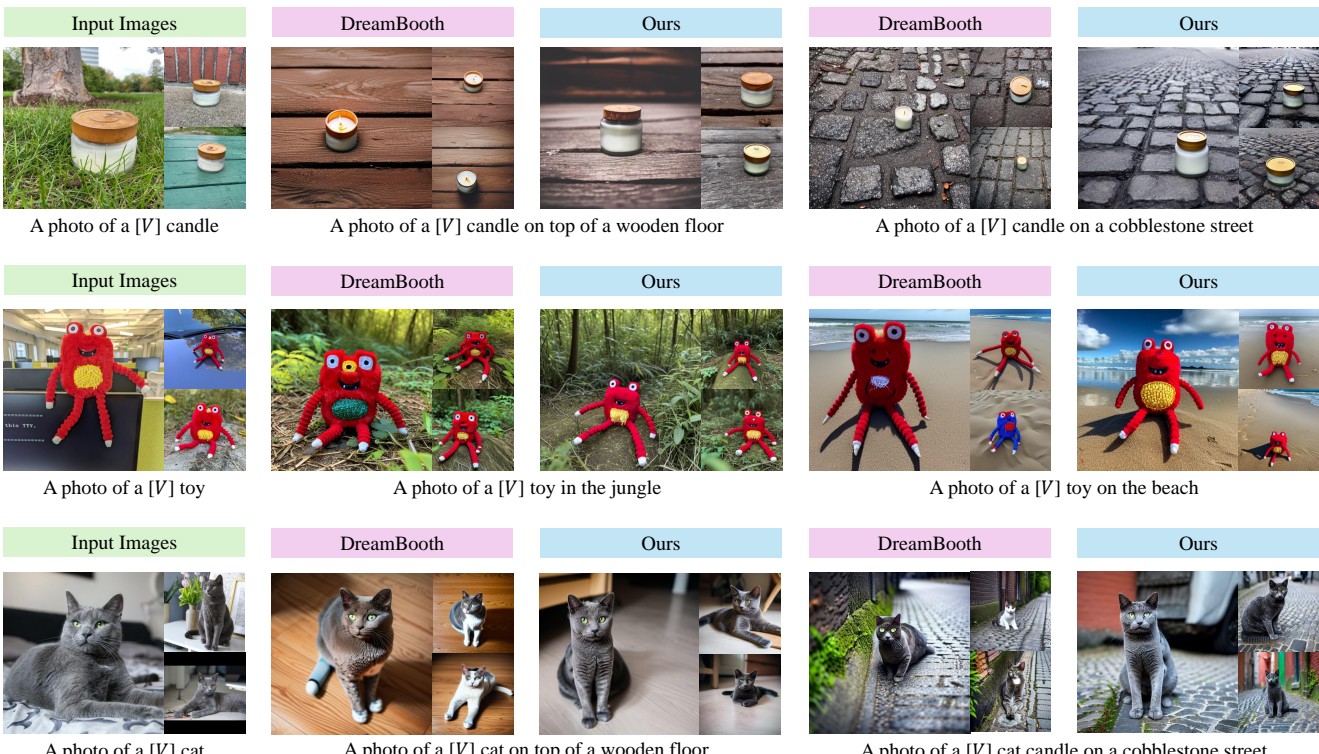

Figure 3. Generated images and their corresponding prompts for both DreamBooth and our method are presented. As shown, our method demonstrates a significantly better ability to preserve the subject's identity, maintaining consistent and recognizable features throughout the synthesized images. This results in higher fidelity and more accurate representations of the subject compared to DreamBooth, which often fails to retain essential details and features. The superior performance of our approach highlights its effectiveness in subject-driven text-to-image synthesis, ensuring both identity preservation and visual coherence.

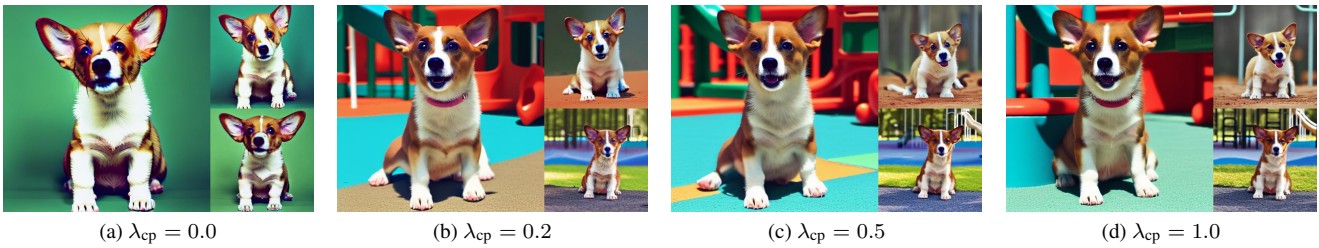

Figure 4. Ablation study for $\lambda_{\text{cp}}$. Images generated by different $\lambda_{\text{cp}}$ with prompt "A photo of a $[V]$ dog on the playground".

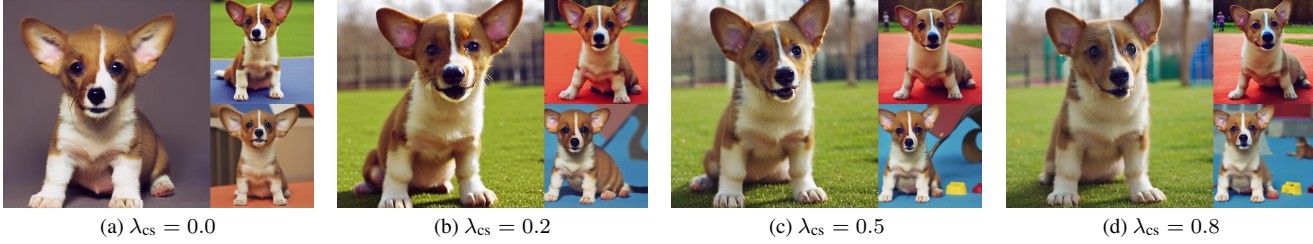

Figure 5. Ablation study for $\lambda_{\text{cs}}$. Images generated by different $\lambda_{\text{cs}}$ with prompt "A photo of a $[V]$ dog on the playground".

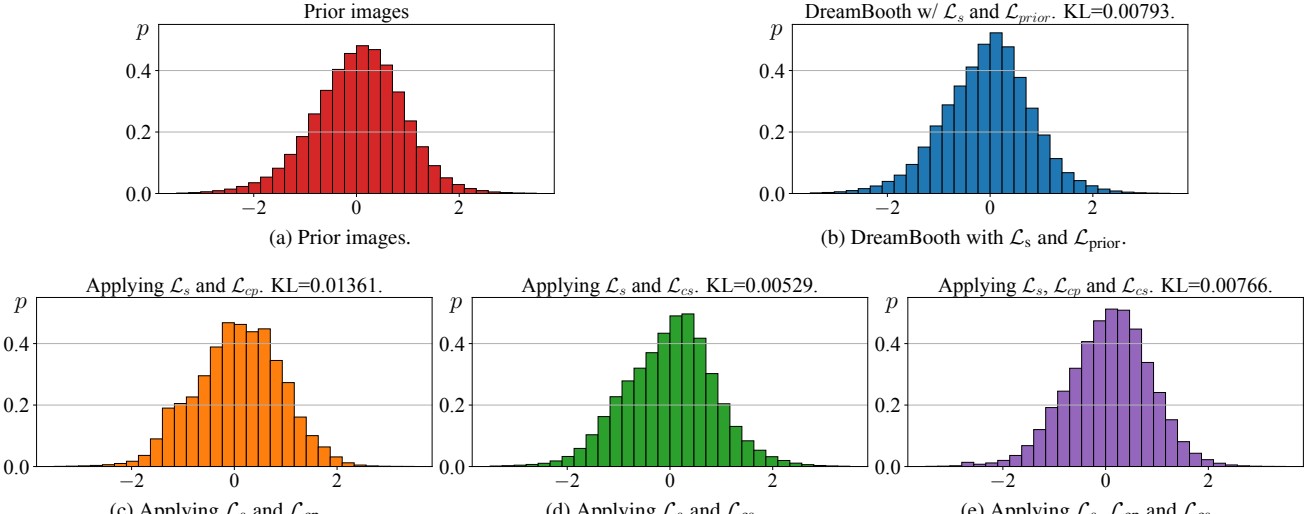

Figure 6. Histogram of the latent codes for generated and prior images under different consistency losses. (a) shows the distribution of prior images, while (c) demonstrates that applying only prior consistency regularization ($\mathcal{L}_{cp}$) shifts the distribution. Our subject consistency loss with multiplicative noise ($\mathcal{L}_{cs}$) in (d) better aligns with the prior distribution. (e) illustrates that combining both regularizations achieves lower KL divergence than DreamBototh in (b), highlighting the effectiveness of our method. The $x$-axis represents latent code values and the $y$-axis represents density $p$.

Booth. This demonstrates that $\mathcal{L}_{cs}$ helps preserve diversity, leading to overall better performance.

**Additive noise *vs*. multiplicative noise.** To justify our use of multiplicative noise over additive noise, we replace multiplicative noise with additive noise and analyze the generated images (Figure 7). The results show that additive noise reduces fidelity and fails to preserve subject identity, whereas multiplicative noise maintains fidelity and preserves subject identity. This occurs because additive noise conflicts with the diffusion process, which also relies on additive noise, making it difficult for the fine-tuned diffusion model to learn subject identity. In contrast, multiplicative noise does not interfere with the diffusion process. Instead, it diversifies the latent code patterns while preserving semantics, effectively maintaining subject identity and enhancing background diversity.

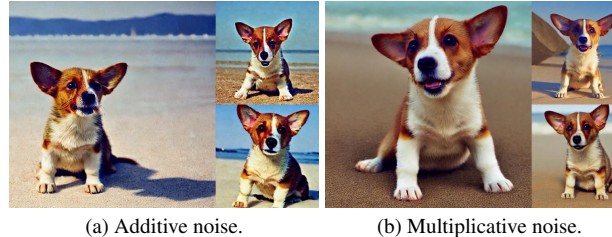

(a) Additive noise.    (b) Multiplicative noise.

Figure 7. Consistency regularization applied to (a) additive noise modulation and (b) multiplicative noise modulation using the prompt *"A photo of a $[V]$ dog on the beach"*.

## 6. Limitations and Future work

While the proposed consistency regularization improves the diversity and fidelity of subject-driven synthesis, it approximately doubles the computational cost in terms of FLOPs and GPU memory during training, as both subject and prior images are processed by the model twice. Inference, however, remains unaffected. In future work, we will explore strategies to reduce training overhead, such as applying consistency regularization intermittently or at intermediate layers. We also plan to evaluate these approaches on other Stable Diffusion fine-tuning tasks, including IP-Adapter [62], ControlNet [64] and Textual Inversion [14].

## 7. Conclusions

To enhance both subject identity preservation and image diversity, our approach introduces two distinct consistency regularizations. First, we enforce consistency between the fine-tuned model's predictions for clean latent codes and those for noise-modulated latent codes of the subject images. Second, we require consistency between the fine-tuned model and the pre-trained model on prior images. These regularizations not only allow the fine-tuned model to better capture and preserve the subject's unique identity, but they also significantly enhance the diversity of generated images. Experiments on subject-driven text-to-image synthesis demonstrate the effectiveness of our method, as it outperforms two strong baselines, yielding superior scores and producing high-fidelity images that more accurately reflect the subject's identity.

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
