# OpenReview forum: "Noise Consistency Regularization for Improved Subject-Driven Image Synthesis"
_thecvf.com/CVPR/2025/Workshop/SyntaGen — SyntaGen 2025 Poster_

### Official Review · Reviewer_qHzx · 2025-03-25

**Rating:** 6
**Confidence:** 3

**Review:**

**Summary**

The paper addresses the challenge of subject-driven personalization in text-to-image synthesis, specifically focusing on the preservation of identity and diversity. It highlights limitations in existing methods, such as DreamBooth, particularly the issues of catastrophic forgetting and reduced image diversity during fine-tuning. To mitigate these, the paper proposes two novel consistency regularizations:

1. Prior Consistency Regularization ($L_{cp}$): This ensures consistency between the fine-tuned and pre-trained models by preserving the class prior information in generated images.
2. Subject Consistency Regularization ($L_{cs}$): This enforces consistency between predictions for clean and noise-modulated latent codes of subject images, enhancing background diversity while maintaining subject identity.

**Strengths**

The introduction of $L_{cp}$ and $L_{cs}$ effectively addresses critical challenges in fine-tuning, offering a theoretical and practical solution to balance subject fidelity and diversity. The use of multiplicative noise modulation is a technical innovation, justified by its effectiveness in preserving identity. Additionally, the paper includes ablation studies that explore various configurations (e.g., varying $λ_{cp}$ and $λ_{cs}$, and testing different noise types), providing valuable insights. Qualitative results, such as visual examples and KL divergence analysis, complement and strengthen the quantitative findings.

**Weaknesses**

1. Computational Resource Intensity: Additional regularization terms likely requires significant computational resources
2. Lack of comparison with other work like Textual Inversion, IP-Adapter,...

---

### Official Review · Reviewer_SRJD · 2025-03-27
**Review for paper: Noise Consistency Regularization for Improved Subject-Driven Image Synthesis**

**Rating:** 6
**Confidence:** 3

**Review:**

Summary:

The paper aims to solve the two key issues in subject-driven image synthesis: underfitting and overfitting. To address the above issues, the paper proposes two auxiliary consistency losses for diffusion fine-tuning: 1. a prior consistency regularization loss ensures that the predicted diffusion noise for prior images remains consistent with that of the pre-trained model. 2. a subject consistency regularization loss enhances the fine-tuned model’s robustness to multiplicative noise modulated latent code, helping to preserve subject identity while improving diversity.

Strengths:

1. The motivation is clear and the methodology is illustrated clearly through Figure. 2.

2. Experiment results demonstrate the effectiveness of the proposed two loss terms.


Weakness:

1. Compared to DreamBooth, the proposed method needs to maintain two model during training: a pre-trained latent diffusion model and a fine-tuned latent diffusion model, which introduces additional cost.

---

### Official Review · Reviewer_T9bj · 2025-03-27

**Rating:** 6
**Confidence:** 4

**Review:**

**Summary**

The paper introduces a method to enhance subject-driven image synthesis using Stable Diffusion. It addresses overfitting and underfitting in fine-tuning by proposing two regularization losses: prior consistency (aligning predictions with the pre-trained model on non-subject regions) and subject consistency (applying multiplicative noise to latent codes to preserve subject identity while improving diversity).


**Strengths**

1. The method is straightforward and relevant to tackle the proposed problem.
2. Experiments include quantitative metrics (CLIP-I, CLIP-T, DINO), qualitative comparisons, and ablation studies validating the impact of each component.


**Weaknesses**

1. Additional analysis and comparison with methods that involve adapters such as IP-Adapter or ControlNet should be added. Since these methods do not modify much the loss function, I wonder if these methods also struggle from the same problem, and if the proposed loss combination can be used to enhance them.
2. Computational Overheads: Training time, resource requirements, and inference speed are not discussed.

---

### Decision · Program_Chairs · 2025-03-30

**Decision:**

Accept (Poster)

**Comment:**

This paper received three borderline acceptances from the reviewers. They praised the clarity of the writing and the comprehensiveness of the experiments. In particular, the introduction of multiplicative noise was highlighted as a key contribution, helping to reduce overfitting to the reference image during personalization. The authors are recommended to add more results compared to other baselines.